# Atlas-scale single-cell multi-sample multi-condition data integration using scMerge2

Yingxin Lin [1,2,3,4], Yue Cao[1,2,3,4], Elijah Willie[1], Ellis Patrick [1,3,4,5] & Jean Y. H. Yang [1,2,3,4] ✉

The recent emergence of multi-sample multi-condition single-cell multi-cohort studies allows researchers to investigate different cell states. The effective integration of multiple large-cohort studies promises biological insights into cells under different conditions that individual studies cannot provide. Here, we present scMerge2, a scalable algorithm that allows data integration of atlas-scale multi-sample multi-condition single-cell studies. We have generalized scMerge2 to enable the merging of millions of cells from single-cell studies generated by various single-cell technologies. Using a large COVID-19 data collection with over five million cells from 1000+ individuals, we demonstrate that scMerge2 enables multi-sample multi-condition scRNA-seq data integration from multiple cohorts and reveals signatures derived from cell-type expression that are more accurate in discriminating disease progression. Further, we demonstrate that scMerge2 can remove dataset variability in CyTOF, imaging mass cytometry and CITE-seq experiments, demonstrating its applicability to a broad spectrum of single-cell profiling technologies.

Technological advances of large-scale single-cell profiling of genes and proteins, such as single-cell RNA-seq (scRNA-seq)[1], Cytometry by Time-Of-Flight (CyTOF)[2] and imaging mass cytometry[3] have exploded in recent years and enabled unprecedented insight into the identity and function of individual cells. This has enabled the discovery of cell-type-specific knowledge and has transformed our understanding of biological systems. This myriad of single-cell data has prompted the recent creation of data atlases that collate single-cell omics data from multiple studies. Examples of large-scale atlases containing over two millions cells are the Human Cell Atlas which aims to map every cell type in the human body[4]; atlas of gene expression and chromatin accessibility of 4 million human fetal cells across 15 organs[5,6]; the Human Tumour Atlas Network[7] and DISCO[8], which provides integrated human single-cell omics data across 107 tissues/cell lines/organoids and 158 diseases. These atlases serve as valuable references for the exploration of healthy and diseased cells.

As single-cell technologies advance, there are an increasing number of studies around the globe that perform multi-condition and multi-sample large-cohort single-cell profiling to examine persisting questions associated with human health. These datasets enable researchers to delve into biological insights of cells under multiple treatment conditions across multiple individuals. For example, to investigate the cell-type-specific cellular mechanism underlying COVID-19 disease severity[9] and to predict treatment response to cancer[10]. Such data and studies are expected to rise in the coming years[11] in the continuing quest to improve human health. This expected increase necessitates the effective access and joint interpretation of multiple datasets to unleash the power of meta-analysis at single-cell resolution.

Last year, benchmarking studies[12] began to investigate atlas-scale integration. Luecken and colleagues investigated 16 popular data integration technologies on 13 data integration tasks with up to 1

[1]Sydney Precision Data Science Centre, The University of Sydney, Sydney, NSW, Australia. [2]Charles Perkins Centre, The University of Sydney, Sydney, NSW, Australia. [3]School of Mathematics and Statistics, The University of Sydney, Sydney, NSW, Australia. [4]Laboratory of Data Discovery for Health Limited (D24H), Science Park, Hong Kong SAR, China. [5]The Westmead Institute for Medical Research, The University of Sydney, Sydney, NSW 2006, Australia. ✉e-mail: jean.yang@sydney.edu.au

million cells. While significant progress has been achieved in batch correction and data integration over the years (including our research), the increasing scale of cohort sizes and the number of related studies for integration has introduced additional scalability challenges. The new challenge for atlas-scale integration is to have a scalable algorithm that can handle a large number of studies, consisting a large collection samples (thousands) and millions of cells. With the exception of Seurat[13], SAUCIE[14] and Scanorama[15], several of these rapid procedures (deepMNN[16], BBKNN[17], Harmony[18], scVI[19], scANVI[20] and DESC[21]) focus on extracting the joint embedding and do not return adjusted gene expression matrices. With the growing need for sample level analysis, the lack of adjusted expression matrices restricts the utilisation of such integrative results and diminishes their potency and generalizability. As a result, the next generation of atlas-scale integration algorithms should be capable of integrating a large number of studies and producing consensus cell type maps as well as adjusted expression matrix for further downstream analysis. In particular, these methods need to overcome the computational challenge of integrating over a million cells and create adjusted gene expression matrix for all genes for downstream analysis.

To this end, we present scMerge2, a scalable, high-capacity algorithm that allows data integration of atlas-scale multi-sample multi-condition single-cell studies. We achieve this through three key innovations in (i) hierarchical integration to capture both local and global variation between studies; (ii) pseudo-bulk construction to ensure computational scalability; and (iii) pseudo-replication inside each condition to capture signals from multiple conditions. Our scMerge2 algorithm is able to integrate many millions of cells from single-cell studies generated from various single-cell technologies, including scRNA-seq, CyTOF, and imaging mass cytometry. Leveraging pseudo-bulk to perform factor analysis of stably expressed genes and pseudoreplicates, scMerge2 is able to integrate five million cells from a large COVID-19 data collection with over 1000 samples from 20 studies globally within a day. We further demonstrate that the integration using scMerge2 improves the performance of discriminating distinct cell states in COVID-19 patients with varying degrees of severity and facilitates diverse single-cell downstream analyses.

## Results

### scMerge2 effectively integrates single-cell multi-sample, multi-condition data

scMerge2 provides a scalable data integration method for the rapid growth of multi-sample, multi-condition single-cell studies. This new extension of scMerge is specifically designed to address unwanted intra- and inter-dataset variation that can overshadow true biological signals between conditions. In our previous study, we introduced scMerge, an algorithm that integrates multiple single-cell RNA-seq data by factor analysis of stably expressed genes and pseudo-replicates across datasets and enhances biological discovery, including inferring cell development trajectories[22]. The integration approach supports diverse integration settings, enabling cross-batch, cross-dataset, and cross-species discoveries. In particular, the semi-supervised aspect of scMerge allows incorporation of prior knowledge facilitated by experimental design.

With the rapid emergence of multi-sample multi-condition single-cell studies and the increased number of datasets for integration, our proposed scMerge2 addresses challenges associated with scalability of cells and studies as well as producing analytically ready data (i.e. adjusted expression matrix). This is achieved via three key innovations as illustrated in Fig. 1. First, hierarchical integration is used to capture both local and global variation. This is a clear contrast to the conventional data integration that involves estimating unwanted variation across all datasets as a whole. When integrating across a large collection (over 10) of datasets with different pairwise differences, sequential integration better captures the difference in pairwise variations.

Two other methods, Seurat and fastMNN, also allow user-defined merging order for data integration tasks. However, these methods require a rigid merging strategy, allowing only pairwise merging at each level and performing batch merging in a progressive manner. In contrast, scMerge2 provides users with a more flexible and adaptable multi-level merging structure, of which each level can comprise multiple collections of several batches and batch correction can be performed within each collection separately using user-defined batch labels. Second, pseudo-bulk construction is used to reduce computing load, allowing for the analysis of datasets containing millions of cells. Third, pseudo-replication inside each condition is built, allowing for the modelling of numerous conditions. Details of these components are included in Methods. In essence, scMerge2 takes gene expression matrices from a collection of datasets and integrates them in a hierarchical manner. The final output of scMerge2 is a single adjusted expression matrix with all input data matrices merged and ready for downstream analysis.

### scMerge2 outperforms existing integration methods in detecting differential expression

We demonstrate the performance of scMerge2 in removing multi-level unwanted variation of multiple scRNA-seq datasets from three aspects. Firstly, to illustrate the effectiveness of the hierarchical integration strategy, we applied scMerge2 to a 200k subset of cells from two COVID-19 studies (Liu and Stephenson) that contain three cohorts/batches within each dataset (See "Methods" for details). We compared the performance of two different scMerge2 settings: scMerge2-h, where we performed intra-study correction before inter-study correction; and scMerge2, where we integrated two datasets (6 batches) in one go. We find that integrating the two studies in a hierarchical manner improves the performance of data integration, especially in terms of revealing the cell type signals (Fig. 2a, b). Compared to the other data integration methods (Seurat, SeuratRPCA, fastMNN, Liger, Harmony, scVI and Scanorama), both settings of scMerge2 (scMerge2-h and scMerge2) have

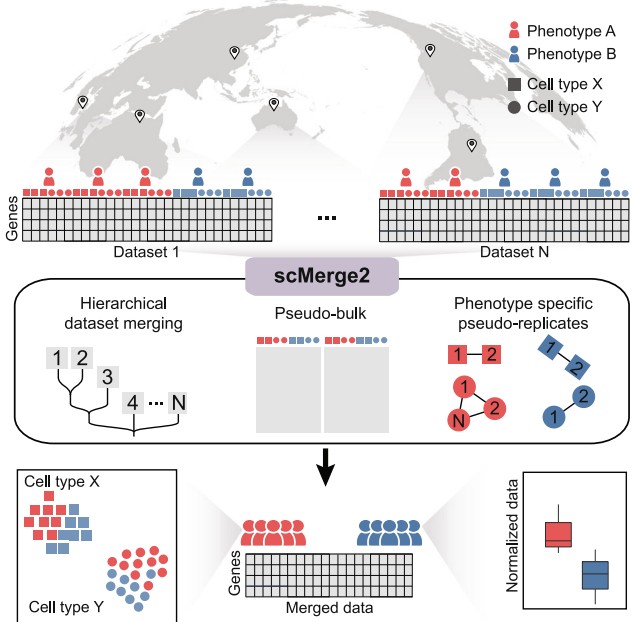

**Fig. 1 | Overview of scMerge2.** This new scalable algorithm uses (i) hierarchical integration to capture both local and global variation; (ii) pseudo-bulk construction to reduce computational load; and (iii) phenotype specific pseudo-replicate, and outputs adjusted expression matrix for millions of cells ready for downstream analysis.

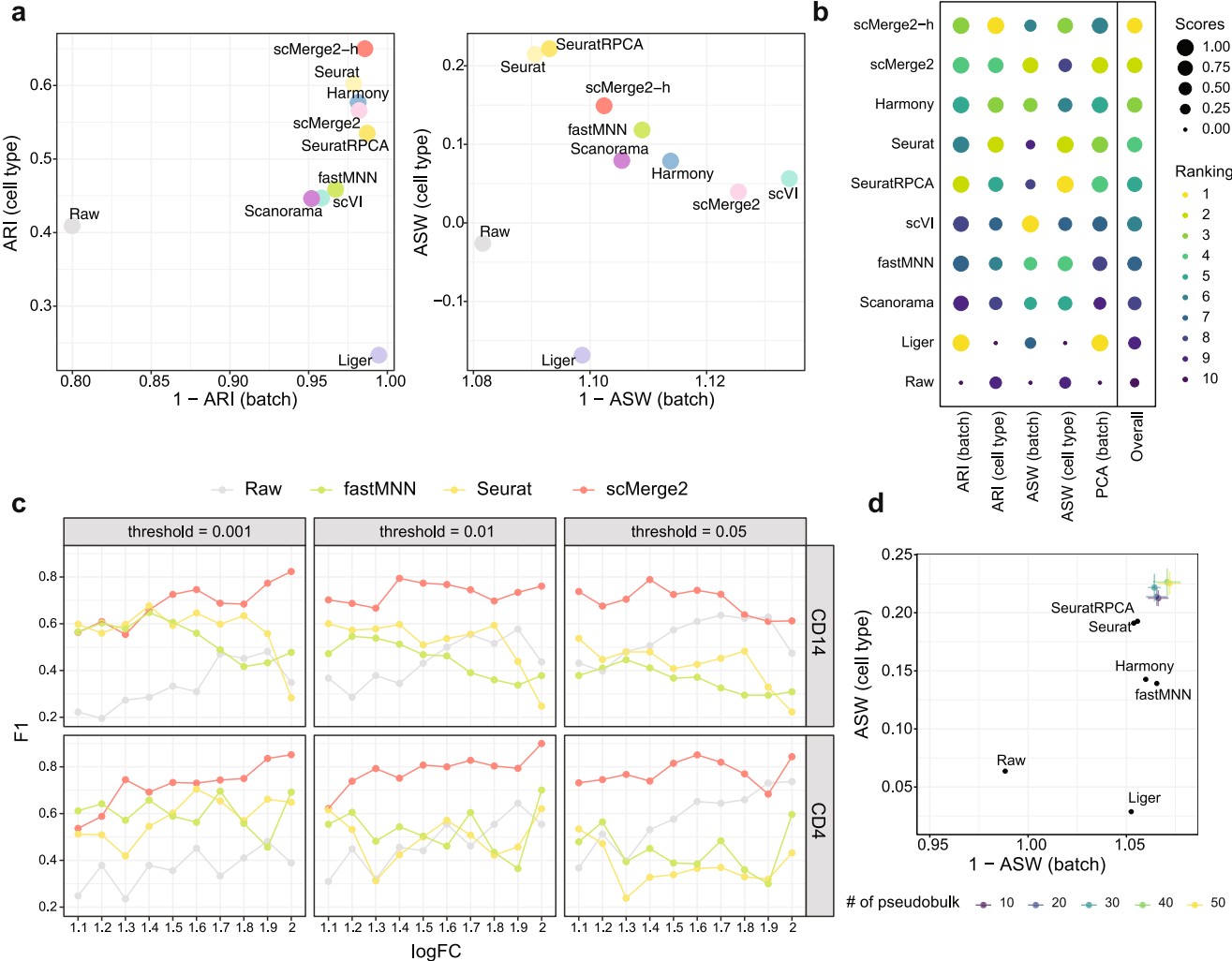

**Fig. 2 | scMerge2 outperforms existing integration methods. a** Scatter plots of evaluation metrics of data integration of a 200k cells subset of two COVID-19 studies (Liu and Stephenson) for scMerge2, scMerge2-h (data merged in a hierarchical manner), Seurat, Seurat (RPCA), Harmony, fastMNN, Liger, scVI, Scanorama and Raw: Adjusted rand index (ARI) (left panel), where x-axis indicates 1 minus batch ARI and y-axis indicates cell type ARI; Average silhouette width (ASW), where the x-axis is 1 minus batch ASW and y-axis is the cell type ASW (right panel). **b** Dot plots indicate the ranking of the data integration methods in terms of five different evaluation metrics. The size of the dot indicates the scaled scores, which are obtained from the min-max scaling of the original values. The overall ranking is ranked based on the average ranking of the five evaluation metrics. **c** F1-score of the differential state (DS) results of two selected cell types (CD14 and CD4) (row) of simulated data, with 10% DS genes within each cell type, for scMerge2, Seurat, fastMNN and raw, varying simulated log fold change (logFC) of DS genes (x-axis) and different threshold of adjusted p-value (column). **d** Scatter plots of evaluation metrics of robustness analysis when varying the number of pseudobulk constructed within each cell type of each batch, where the x-axis is 1 minus batch ASW and y-axis is the cell type ASW. Source data are provided as a Source Data file.

overall better performance in achieving the balance of batch effect removal and biological signal preservation, based on the five evaluation metrics that quantify the data integration performance (Fig. 2a, b). We further performed the comparison with fastMNN and Seurat, specifying the merging order to correct the cohort effect within the study first, and then corrected the study effect (fastMNN-h and SeuratRPCA-h). We demonstrate that scMerge2h consistently outperforms the other two methods in terms of overall performance (Suppl. Fig. S1)

In terms of computational efficiency and memory required, we evaluated scMerge2 on data integration tasks using the full set of the two COVID studies (Liu and Stephenson), which in total have ~1M cells. We benchmarked the scalability of the integration methods by varying the number of cells and the number of features (~17k genes or 87 proteins). We find that among all the methods that are able to return the adjusted gene expression matrix, scMerge2 is the most efficient in terms of computational time in all comparisons (Suppl. Figs. S2–S3). In terms of memory usage, to return a fully adjusted gene expression matrix, scMerge2 requires more memory than fastMNN and scVI, but significantly less than Seurat.

Next, we investigate the performance of the adjusted matrix in identifying genes that are differentially expressed between two conditions (termed as differential state (DS) analysis by ref. 23) through a simulation study. We generated synthetic single-cell datasets with two batches and multiple samples from two conditions using a simulation framework that extended from scDesign3 model[24], with known ground truth DS genes (Suppl. Figs. S4–S5) (See Methods). Cell-type-specific DS analysis was performed using the limma-trend algorithm[25] on the sample-wise aggregated data by taking the mean of the log-transformed or adjusted data. By simulating data with different log fold change (1.1 - 2) and proportions of DS genes (5% and 10%), we find that scMerge2 substantially outperforms the other two data integration methods that also return adjusted matrices in detecting DS genes (Fig. 2c and Suppl. Fig. S6). scMerge2 has much lower FDR than fastMNN and Seurat, and higher TPR compared to the unadjusted data (Suppl. Figs. S7–S8), illustrating that scMerge2 outputs an adjusted

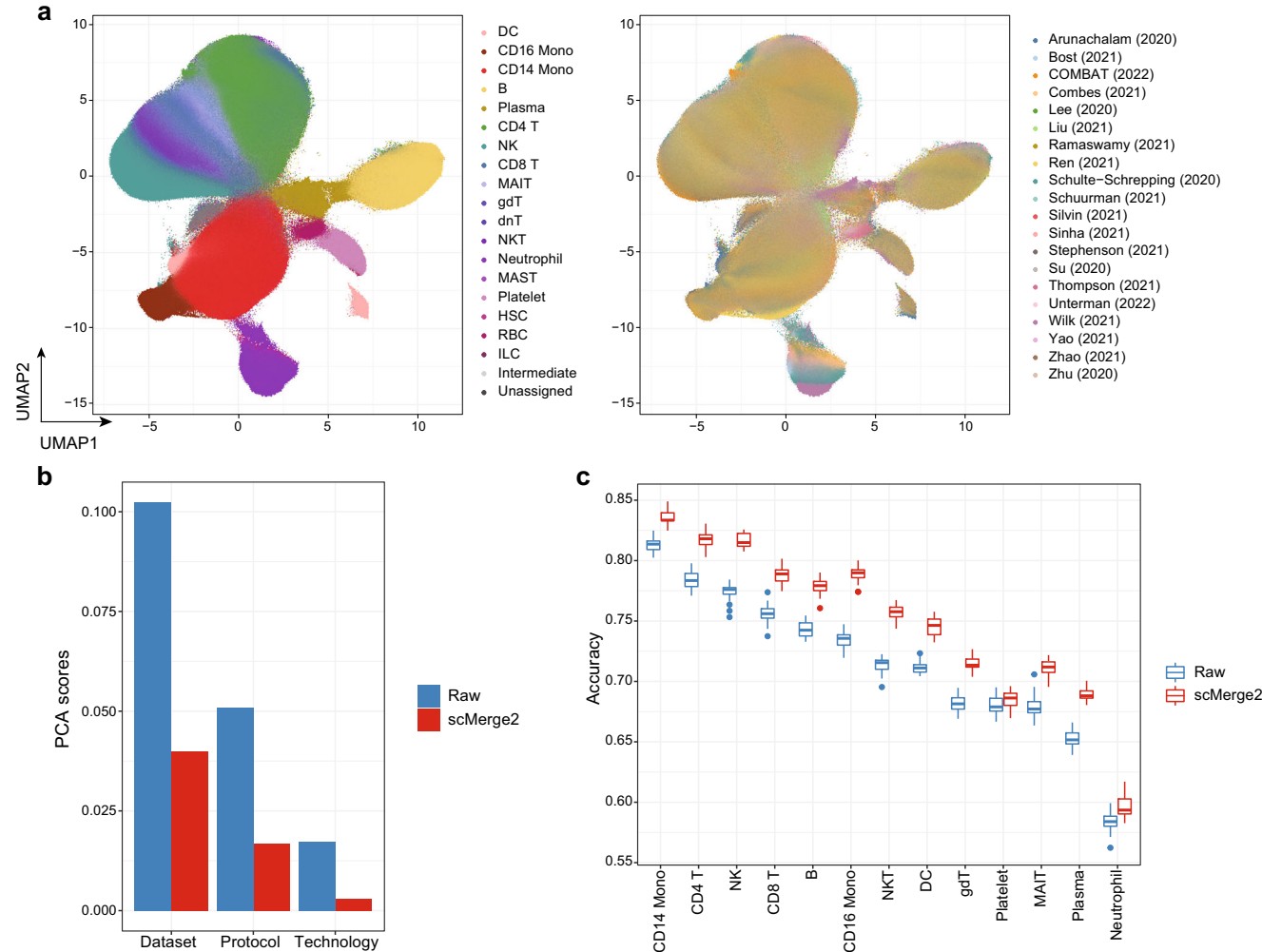

**Fig. 3 | scMerge2 is scalable to integrate five millions COVID-19 PBMC cells.**
**a** UMAP of integration of COVID-19 data collection by scMerge2, coloured by cell type (left) and studies (right). **b** Evaluation metrics of PCA scores using dataset, protocol and technology as labels, comparing raw logcounts (blue) and scMerge2 normalised results. A lower score indicates better unwanted technical variation removal. **c** Prediction results from 20 times repeated cross validation of disease severity using cell type-specific aggregated expression calculated from raw logcounts (blue) and scMerge2 normalised results (red). Each box includes 20 points, ranges from the first to third quartile of classification accuracy with the median as the horizontal line. The box plot's lower whisker extends 1.5 times the interquartile range below the first quartile, while the upper whisker extends 1.5 times the interquartile range above the third quartile. Source data are provided as a Source Data file.

matrix with less unwanted variation for single-cell downstream analysis.

Finally, we illustrate the robustness of scMerge2 by varying the key tuning parameters of the algorithm, including the number of unwanted variation factors, the number of pseudo-bulk, the ways of pseudo-bulk construction and the number of nearest neighbours. As shown in Fig. 2d and Suppl. Figs. S9, despite varying the settings in the algorithm, scMerge2 has consistently better performance than the other methods. Together, these results demonstrate the effectiveness, utility and computational efficiency of scMerge2 in data integration of scRNA-seq data.

## scMerge2 is scalable to integrate five millions COVID-19 PMBC cells

To demonstrate the scalability of scMerge2 in integrating multi-sample multi-condition single-cell data, we performed scMerge2 on a COVID-19 data collection of consisting of ~5m cells from 1298 samples (963 individuals) PBMC samples from 20 studies worldwide (See Methods). We considered the cell type annotation refined by scClassify as pseudo-replicates information. We also used a hierarchical integration strategy, where we first performed integration of different cohorts within one study respectively (e.g. Ren, Stephenson, Liu and Schulte-Schrepping) and also two studies with distinguished sequencing depth, followed by the integration of 13 studies with small number of cells (hierarchical integration strategy shown in Suppl. Fig. S10). We then integrated all the data in the next step. An inspection of UMAP visualisations shows that scMerge2 effectively integrates the 20 studies, while preserving the multi-level cell type information (Fig. 3a, Suppl. Fig. S11). A UMAP plot faceted by dataset further illustrates the successful removal of dataset induced unwanted variation (Suppl. Fig. S12). The quantitative evaluation metrics further confirm this observation, where we find that scMerge2 reduces the technical variation caused by dataset, protocol and technology, resulting in improved cell type identification (Fig. 3b, Suppl. Fig. S13).

To further illustrate the utility of scMerge2, we demonstrate that it improves the prediction of disease severity in the COVID-19 dataset using cell-type-specific expression. Comparing to the original raw log-normalised data, identifying cell types with scMerge2 substantially improves the prediction accuracy rate of disease severity for all cell types that have more than 1% abundance in the data, with a 3.2% increase in accuracy on average (Fig. 3c and Suppl. Fig. S14). Notably, we find that CD14 Monocytes have the highest discriminative power

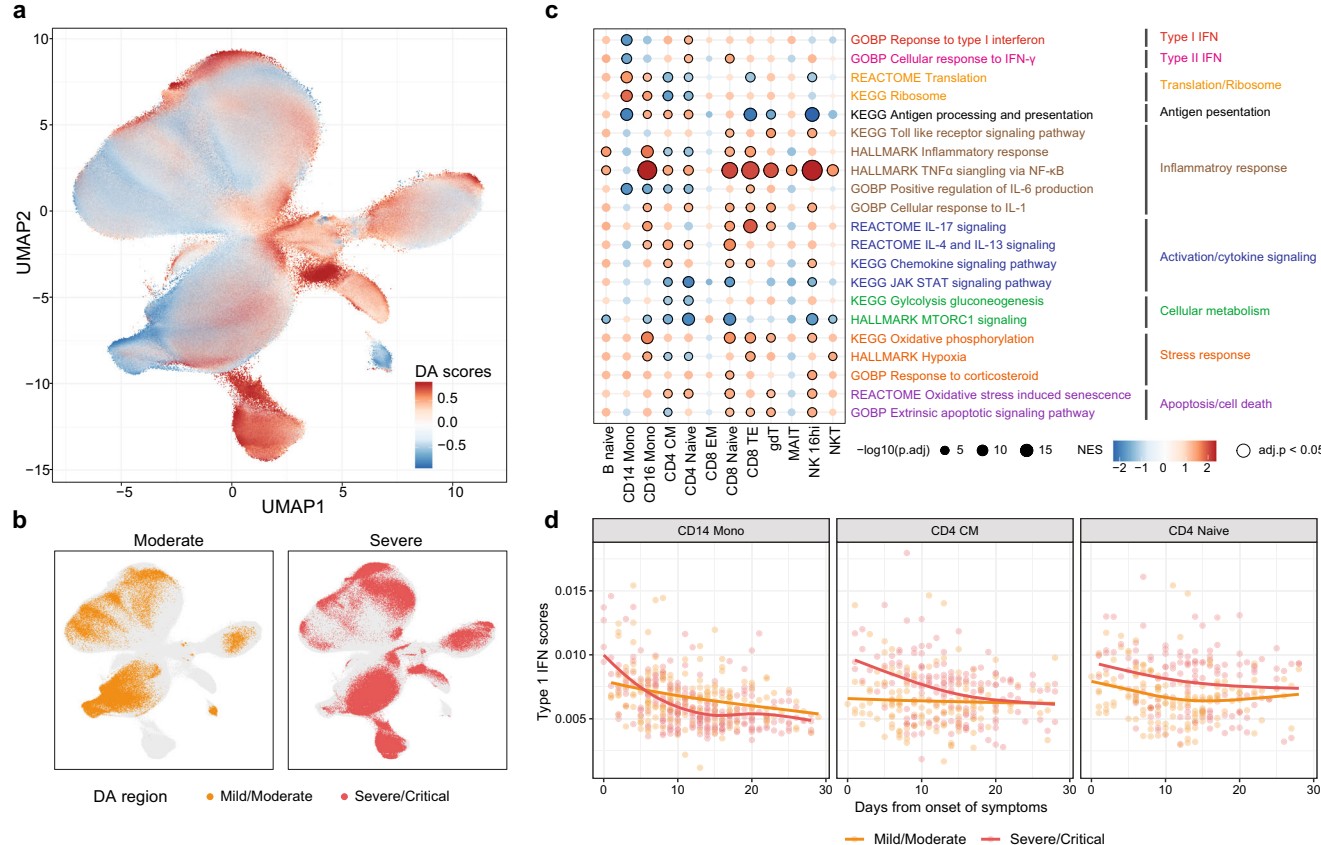

**Fig. 4 | scMerge2 enables differential cell state detection for multi-condition data. a, b** UMAP plot of integrated COVID-19 data coloured by (**a**) differential abundance (DA) probability scores calculated by DA-seq between the moderate and severe patients, where higher scores indicated the cells are more related to severe states; **b** DA region associated with disease severity identified by DA-seq. **c** Enrichment scores of selected pathways for cell-type-specific differential expressed genes distinguished the severity, where a higher score indicates a higher enrichment associated with severe states. The size of the dot indicates the −log10 adjusted p-value, where black circles indicate statistical significance (Benjamini-Hochberg adjusted *p*-value < 0.05) from two-sided gene set enrichment test using R package *fgsea*; and the colour indicates the normalised enrichment scores of the pathways. **d** Scatter plots showing per-sample gene set signatures (Type-1 IFN) calculated from the scMerge2 normalised data along the days since symptom onset, coloured by disease severity of the patient. CD14 Monocytes, CD4 CM and CD4 Naive are shown as examples. Source data are provided as a Source Data file.

for disease severity among all cell types, and scMerge2 is able to further improve the accuracy rate from 81.3 to 83.6%.

## scMerge2 enables differential cell state detection for multi-conditions data

We next illustrate how the adjusted expression matrix output from scMerge2 facilitates several downstream analysis of single-cell multi-condition multi-sample studies, including differential abundance analysis and differential expression analysis. As a case study, we focus on the analysis of identification and characterisation of cell states that are distinguished between the moderate and severe patients using COVID-19 data collection. We first calculated the differential abundance score for each cell to quantify the difference between the moderate and severe patients using DASeq[26]. As shown in Fig. 4a, b, we are able to identify regions on the UMAP plots that are associated with the disease severity. As expected, when mapping these regions to cell types, we find that neutrophils have the highest proportion of cells that are associated with severe disease outcome as their accumulation marks the critical illness of COVID-19 patients[27] (Suppl. Fig. S15).

Next, we investigate the cell-type-specific underlying biological process pathways that are associated with the disease severity and time for each cell type. We performed the differential expression analysis on the cell-type specific pseudo-bulk by considering both disease severity and days from onset of symptoms as covariates, followed by gene set enrichment analysis (GSEA). The pathways enriched

with disease severity include hallmark TNFα signalling and hallmark inflammatory response (Fig. 4c) and are upregulated in severe patients in most of the cell types, while GO IL6 positive production and Hallmark MTORC1 signalling are upregulated in moderate patients. Notably, we observe that a few pathways reveal distinct enrichment patterns between different cell types, including GO response to type-I IFN. We find that for CD14 Monocytes (Fig. 4c, d), the type-I IFN signatures is negatively associated disease severity and also decrease over time, consistent with the previous findings[28] (Fig. 4d). While other cell types such as CD4 CM and CD4 Naive have an enrichment of type-I IFN in severe patients, this enrichment is also decreased over time. Together, these analysis demonstrate that the integration of multiple studies using scMerge2 enables a variety of data analysis approaches that address a wide range of biological questions.

## scMerge2 is versatile to other single-cell platforms

One of the key strengths of scMerge2 is its generalizability to data from multiple biotechnology platforms. We illustrate that scMerge2 is generalizable to other single cell modalities including spatially resolved modality and multi-modalities. We start by illustrating that our algorithm is directly applicable to other single-cell single-modal data, using two mass cytometry time-of-flight (CyTOF) datasets as an example. The two datasets (COMBAT (CyTOF) and Geanon (CyTOF)) contain more than 11 million cells in total collected from healthy controls, COVID-19 and sepsis patients, with 18 immune cell

populations and activation states. The UMAP plots constructed after integration (Fig. 5a) reveal that the two datasets are successfully integrated compared to the raw data. Notably, we find that Granulocytes (Neutrophils and Eosinophils), cell types that are only present in Geanon (CyTOF) but not COMBAT (CyTOF), are represented as a discrete and distinct cluster, suggesting that scMerge2 is able to reveal the unique cell types existing only in specific batches. An inspection of the cell-type-specific marker expression distribution further confirms the effective dataset effect removal (Fig. 5b and Suppl. Fig. S16).

Next, we show that scMerge2 enables normalisation of spatially resolved single-cell data for better cell type identification with specific cluster markers. We applied scMerge2 to a COVID-19 Imaging Mass Cytometry (IMC) dataset[29], followed by clustering using FlowSOM[30], with the number of clusters set equal to the manually annotated cell types in the original study. We find that compared to the original data, the scMerge2 adjusted matrix provides better clustering results that are more consistent with the manual cell type annotation (Fig. 5c), with ARI increasing from 0.13 to 0.58. These clusters are also marked by more specific enrichment of protein markers (Fig. 5d). For example, scMerge2 is able to reveal a cluster of T cells that uniquely expressed CD8a but not CD4 and a cluster that expressed of CD4 but not CD8a. Similarly, scMerge2 identifies the B cell cluster that has high expression in CD20, while clustering directly on the unadjusted matrix results in several clusters with qualitatively similar enrichment of markers, lacking the ability to identify distinguished cell types (Fig. 5e).

Lastly, we demonstrate scMerge2 can efficiently remove the unwanted variation of multi-modal data, such as Cellular Indexing of Transcriptomes and Epitopes by Sequencing (CITE-seq) data that concurrently measure RNA and cell-surface proteins of the same cell. In this case, we can remove the unwanted variation for each of the two modalities separately using scMerge2. We first examined the quality of data integration using two CITE-seq datasets with six batches and 87 common surface proteins measured (The same data used in Fig. 2a, b). We find that scMerge2 utilising the hierarchical merging strategies achieves a better balance between batch effect removal and cell type signal preservation than most of the other methods, with comparable performance with Harmony (Suppl. Fig. S17). Similar to the findings in scRNA-seq, using surface protein expression adjusted by scMerge2 improves the severity prediction, compared to the raw data (Suppl. Fig. S18). With the adjusted expression matrix of each modality, one can perform any multi-modal integration approach to obtain the joint latent space and visualisation of cells with batch effect removal[13,31,32]. As an example, we used j-UMAP that generates joint visualisation of the adjusted multi-modal data[32], which further confirms the effective integration of the six batches from the two CITE-seq datasets (Fig. 5f).

## Discussion

We have presented scMerge2, a scalable approach for integrating data from large-scale multi-sample multi-condition single-cell studies. This was achieved via the use of three essential innovations with hierarchical integration, pseudo-bulk building to minimise processing demand, and pseudo-replication that accounts for circumstances with phenotypes. Our algorithm enabled the atlas-scale integration of 20 global COVID-19 studies with around 5 million cells from 963 donors, 1298 samples. We illustrated that scMerge2 data integration enabled the detection of distinct cell states in COVID-19 patients of variable severity. Finally, scMerge2 merged millions of cells from a number of single-cell technologies, including as CITE-seq, CyTOF, and image mass cytometry.

The type of output extracted from atlas-scale data integration has an important impact on the analytical question of interest. To date, there are three standard types of output from recent atlas-scale data integration (defined as over millions of cells). These are (i) an adjusted gene expression matrix, (ii) a low-dimensional projection of the data, known in machine learning as "embeddings"; and (iii) a unified graph representation. Various methodological approaches may provide one or more of these types of outputs. In general, there are a number of existing approaches that use modern deep learning-based algorithms to achieve fast, atlas-scale integration. Given that single-cell data are ultra sparse high-dimensional datasets, "embeddings" are a natural output since they are effective for joint data visualisation and reduce memory load. However, an embedding output by itself increases interpretability challenges since a low-dimensional representation does not naturally lend itself to the development of interpretable features such as cell–cell interactions or pathway information, which is crucial for downstream case-control studies or multi-treatment analysis. One step towards achieving a balance between generating adjusted expression matrices and appropriate memory usage is to enable selective adjusted output. For example, scMerge2 enables the extraction of a subset of genes (such as the top $n$ highly variable genes) of the adjusted matrix for all 5 million cells in the COVID-19 data sets as well as outputting the adjusted matrix by batches, allowing users to effectively balance computational burden with specific downstream analytical strategies.

The order of integration is an important factor in hierarchical merging, which can be knowledge-guided or data-guided. Our current method is based on a data-guided order, in which we integrate batches within one study or studies with similar size first. In contrast, a priori information such as sequencing platforms or cell extraction techniques can be used in knowledge-guided order of integration. Noted that the hierarchical data integration design can be broadly classified into two strategies[33], balanced trees and concatenating approaches. The balanced tree approach integrates between pairs of datasets at different levels of the tree, and the procedure is continued until all data is merged. The concatenating approach sequentially integrates datasets, therefore for $n$ data sets, this will need $n - 1$ steps of integration. Previous studies have found that normalisation results are very similar between the two types of integration tree structures[33]. The key difference between the approach is computational burden with the concatenating approach being more computational intensive. Currently, the scMerge2 approach is closer to the balance approaches allowing for many datasets to be added simultaneously at each level.

We demonstrated that our curation and effective integration of the COVID-19 gene expression data with over 1000 individual samples facilitates flexible downstream meta-analysis, offering the opportunity to examine particular sub-populations that cannot be adequately addressed with individual datasets. Scientists, for example, may investigate the molecular differences underlying mild and severe outcomes for a given age group (e.g., middle-aged individuals between 41 and 50). Such analyses are difficult to perform in individual studies due to the limited sample sizes. This challenge can be overcome by merging several datasets.

Recent technological advancements substantially extend beyond scRNA-seq, enabling other data modalities (e.g. DNA, proteins) to be profiled in individual cells providing a more comprehensive molecular view of the cellular regulation. For the datasets with multi-modal profiles measured for the same cell (paired data), such as CITE-seq and ASAP-seq, scMerge2 can be applied to integrate data from different batches by either considering each each modality as a separated matrix, or concatenating the data into a single matrix. Currently, the integration illustrated in this paper was done within each modality. In the future, we can incorporate the multi-modal information to better identify the pseudo-replicates of the paired data as well as utilise the higher-order relationship of features to improve the integration performance.

In summary, scMerge2 enables atlas-scale integrative analysis of large collections of single-cell data. As the availability of public multi-sample multi-conditional single-cell studies continues to surge, scMerge2 demonstrates its ability to integrate over 5 million cells for further downstream analysis, thereby enabling effective downstream

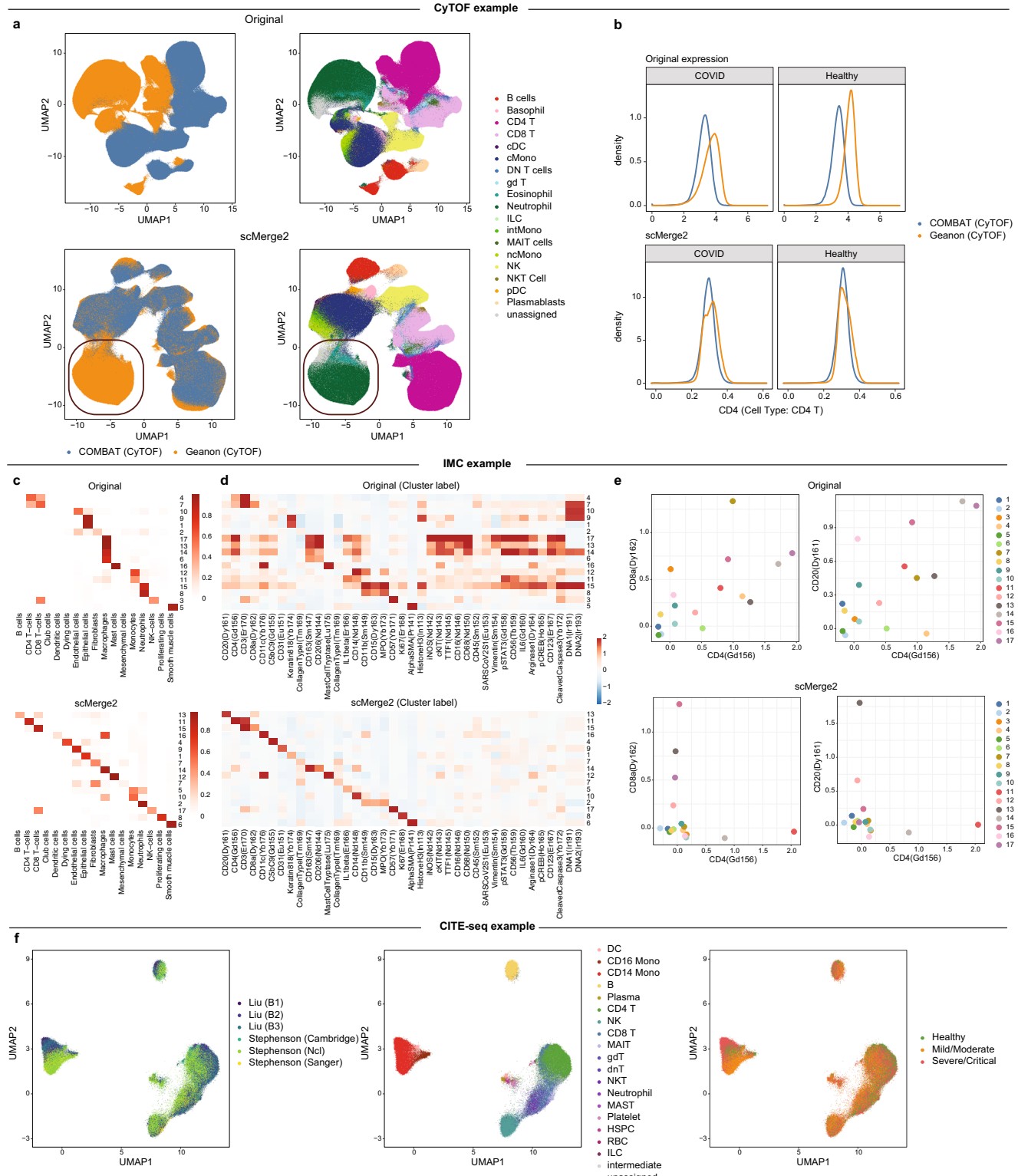

**Fig. 5 | scMerge2 is versatile to other single-cell platforms. a** UMAP plots of CyTOF data coloured by dataset (left) and cell type (right), for original (first row) and scMerge2 (second row). The red circles highlight the cell types (Neutrophils and Eosinophils) that are unique to Geanon (CyTOF). **b** Density plot of selected markers in specific cell types (CD4 in CD4 T cells), using original expression (first row) and scMerge2 adjusted expression (second row). Within a specific cell type, the distribution of the cell type markers are expected to be similar between two datasets. **c** Heatmaps indicate the clustering results and their fractions of concordance with the original cell type annotation given in ref. 29 for Original (first row) and scMerge2 (second row). Clearer diagonal structure illustrates better

concordance. **d** Heatmaps indicate the average marker expression, calculated from cells aggregated by clusters for Original (first row) and scMerge2 (second row). More specific markers for each column and row indicates more distinguished clusters being identified. **e** Scatter plot indicates the average marker expression for each cluster, calculated using Original data (first row) and scMerge2 adjusted data (second row), for two pairs of protein markers: CD4 vs CD8 (first column); and CD4 vs CD20 (second column). Low concordance between the two markers is expected to reveal cluster with specific markers. **f** J-UMAP plot of integrated CITE-seq data coloured by dataset (left) and cell type (middle) and severity (right). Source data are provided as a Source Data file.

meta-analysis. Notability, when compared to the raw log-normalised data from the outset, we demonstrated that scMerge2 offers a significant improvement in the prediction accuracy rate across all of the main cell types. The merge of large collections of scRNA-seq datasets from several cohorts further enables identification of distinct cell states in COVID-19 patients whose symptoms are of varying degrees of severity. Finally, scMerge2 has the ability to combine the data from millions of cells obtained from a variety of single-cell technologies, such as CITE-seq, CyTOF, and image mass cytometry.

## Methods
### scMerge2
**Single-cell grouping within one batch.** Following the same principals as scMerge, the scMerge2 approach begins by grouping the cells that share similar biological signals within each dataset or batch. We can approach this in two ways: one way is to perform unsupervised clustering; the other way is using results from supervised cell type classification.

- Clustering-based grouping: This is performed by default when no cell type label is used as input. Firstly, the top 2000 highly variables genes (HVG) are selected using *getTopHVGs* in the *scran* R Package, using batch information as block information. For data like CyTOF and ADT from CITE-seq data, this step will be skipped and all features will be used in the next step. Next, within each batch, instead of using k-means clustering as in the previous version, we construct a shared nearest neighbour graph on the gene expression of the HVGs, with a default number of neighbours of 10, followed by louvain clustering. This therefore relieves the need of predefining the number of clusters that is required in our previous version.
- Reference-based grouping: This refers to the use of supervised cell type classification to predict or annotate the cell types using one or more reference datasets. This ensures the cell-type annotations are consistent among datasets. Cell type classification algorithms (e.g. scClassify[34] and SingleR[35]) can also be used and the reference dataset can be external datasets with similar cell types to the data to be integrated. This approach unifies cell type annotation across all datasets and eliminates the need for clustering and cell type annotation after data integration. It is noted that this approach is used in the COVID-19 case study to integrate the data collection of 20 datasets.

**Pseudo-bulk construction.** With the cell type grouping of each batch determined, scMerge2 next constructs multiple pseudo-bulk within each cell type. The pseudo-bulk construction significantly reduces the computational time in two main steps of the original version of scMerge[22]: identification of pseudo-replicates and RUVIII model estimation. scMerge2 provides two approaches to calculate cell-type-specific pseudo-bulk for each batch:

- when count data are not available for all datasets, for each cell type grouping, we randomly assign the cells into $k$ subsets and take the gene-wise average of each subset as one pseudo-bulk. This therefore results with $k$ pseudo-bulk for one cell type grouping.
- when counts data are available for all data, we can perform a similar pool-and-divide strategy that is proposed in RUVIII-NB[36]. Here, we can have two strategies in pooling the cells: (1) assign the cells based on library size; (2) randomly assign the cells into $k$ subsets. Then we gene-wisely take the sum of the counts for each subset and generate the counts data following a negative binomial distribution. While the pseudobulk matrix generated by this strategy is able to maintain the gene mean-variance relationship[36], we find that this approach does not improve the quality of data integration in scMerge2 (Suppl. Fig. S9).

Noted that $k$ is set as 30 by default for cell type group with more than $k$ number of cells, and pseudo-bulk are not constructed for cell types with less than $k$ cells, i.e., all the cells from these cell types will be retain for the next steps of scMerge2.

**Pseudo-replicates identification across batches in scMerge2.** Replicates are considered as the samples with similar biological variation across batches. Construction of pseudo-replicates is one of the key steps in scMerge which later are utilised to estimate the unwanted variation from the data. In scMerge, we proposed a five-step procedure to identify pseudo-replicates by clustering on a mutual nearest cluster (MNC) graph, where each node of the MNC graph indicates a group of cells in a batch. scMerge2 follows similar steps as the previous version, but with two major improvements:

- The pseudo-replicates identification is based on the pseudo-bulk matrix to reduce the computational time;
- For data with multiple conditions (or other observed biological factors), scMerge2 allows the MNC graph to be constructed within each condition to preserve the biological variation. Note that this strategy can only be used when the batches to be merged have at least one common condition and can only be performed in the condition with multiple batches.

**Estimation of RUVIII model using pseudo-bulk.** The underlying model of scMerge2 is the fastRUVIII model that takes the gene-wise standardized gene expression matrix that is log-transformed and cosine normalised as input. Let $Z_{cg}$ be the standardized data, where $c = 1, ..., C$, with $C$ indicates the number of cells from all batches/datasets in total; $g = 1, ..., G$, with $G$ indicates the number of genes. Following the same annotation in scMerge, we formulate $Z_{C \times G}$ using RUVIII model as

$$Z_{C \times G} = X_{C \times p}\beta_{p \times G} + W_{C \times k}\alpha_{k \times G} + \epsilon_{C \times G}, \qquad (1)$$

where $X$ denotes the matrix of observed factors of interest; $p$ denotes the number of factors of interest; $W$ denotes the matrix of unobserved factors of unwanted variation; $\alpha$ denotes the coefficient of $W$; $k$ denotes the number of unwanted factors, which is unknown (set as 20 by default for scRNA-seq data, and 10 for ADT from CITE-seq data and CyTOF data); $\epsilon$ denotes the random error. Following the RUVIII model estimation proposed in refs. 22,37, the model removes the unwanted variation from $Z_{C \times G}$. In summary, it follows the three steps:

- Step i: estimate $\alpha$ via the first $k$ right singular vectors of Singular Value Decomposition (SVD) on $R_M Z$, where $R_M = 1 - M(M^T M)^{-1}M^T$, with the replicate matrix $M \in R^{C \times N}$, $N$ indicates the number of types of pseudo-replicates;
- Step ii: estimate $W$ by $W_{C \times k} = Z_s \hat{\alpha}_s^T(\hat{\alpha}_s \hat{\alpha}_s^T)^{-1}$, where $\hat{\alpha}_s \in R^{k \times G_s}$ indicates the the submatrix of $\alpha$, which columns include only the genes that belongs to single-cell stably expressed genes (SEG) with number of genes as $G_s$ (SEG selection and evaluation can be found in ref. 38);
- Step iii: adjust the matrix by subtracting the estimated unwanted variation component:

$$\hat{Z}_{C \times G} = Z_{C \times G} - \hat{W}_{C \times k}\hat{\alpha}_{k \times G}. \qquad (2)$$

SVD is a computationally intensive algorithm, especially for large matrices like single-cell data. We argue that for Step 1, we do not need the full single-cell data to estimate $\alpha$. Instead, we can subsample the data or construct cell-type-specific pseudo-bulk which are informative enough to approximate the full single-cell matrix to reduce the computational burden in estimation of $\alpha$. Let $Z_{C_b \times G}$ denote the the "sketch"

of the full single-cell matrix derived from pseudo-bulk construction step, where the column denotes the number of the genes, with the same dimension as the full data $Z$; the row now indicates the number of pseudo-bulk, with dimension $C_b$. We then construct pseudo-replicates based on the pseudo-bulk matrix $Z_b$ to obtain the replicate matrix $M_b \in R^{C_b \times N_b}$ (See Section *Pseudo-replicates identification across batches in scMerge2* for more details). We estimate $\hat{\alpha}^b$ using the first $k$ right singular vectors of SVD on $R_{M_b} Z_b$. By treating $\hat{\alpha}^b$ as the approximation of $\hat{\alpha}$, we then next bring back the full single-cell matrix $Z$ to estimate $W$ and adjusted $\hat{Z}$ following the same Steps 2-3 above.

**Hierarchical merging.** When we integrate data from different studies, the unwanted variation can come from multiple levels, such as batch effect of samples within each study but also between studies. In this case, a hierarchical integration strategy would be useful to first adjust intra-study unwanted variation effect, and then perform the inter-study data integration. On the other hand, when we integrate a large number of studies, such as the COVID-19 data collection in this paper, starting from correcting the data of a smaller set of studies can be a more efficient way to estimate the parameters of the model to harmonise the data[33].

scMerge2 allows users to input a hierarchical tree strategy to perform the data adjustment in a multi-level manner. This data strategy can take a flexible multi-level merging structure. For each level, it can contain multiple collections of batches and data correction can be performed within each collection respectively, with a user-defined batch label. For each collection, it can consist of multiple batches. The data adjusted on the current level will be used as input on the next level.

For the COVID-19 200k data collection, we first integrated the the 3 batches within each dataset before integrating the two datasets. For the COVID-19 scRNA-seq data collection, we first performed the adjustment on four datasets that have multiple cohorts (Ren, Stephenson, Liu and Schulte-Schrepping) to correct the intra-study unwated variation (where the cohort label is used as batch label) as well as between the two datasets that have very different sequencing depth (Arunachalam and Wilk). Next, we performed the adjustment of the 13 datasets with <200,000 cells. We finally integrated all the 20 studies together, where the study label is used as batch label.

### Data collection and preprocessing

**COVID-19 scRNA-seq data collection.** We collected 20 public COVID-19 PBMC and whole blood scRNA-seq datasets (Supplementary Table 1). The raw count matrix of each dataset is size-factor standardized and log-transformed using *logNormCount* function from *scater*[39] R package. To unify the cell types from different studies, we performed scClassify to reannotate the cell types based on a 3-level hierarchical cell type tree[34], using three distinct reference datasets that were either generated from whole blood (Wilk) or generated by CITE-seq protocol that contains multi-level annotations (Liu and Stephenson).

**COVID-19 200k CITE-Seq data collection (COVID-19 200k).** To benchmark scMerge2 with other methods, we subset 200k cells from the two COVID-19 studies (Liu and Stephenson) as a benchmarking dataset that with 17,446 genes, 87 proteins and 184 samples from 3 conditions (Healthy, Mild/Moderate, Severe/Critical) to assess the concordance performance of the adjusted gene expression matrix after data integration. Both of these two studies have three batches within the studies, which allows us to evaluate the hierarchical merging strategy in scMerge2 (i.e., scMerge2-h), where we first integrated the three batches within each batch, with $k_{RUV} = 10$ ($k_{RUV}$ denotes the number of unwanted variation) and then performed the integration across two datasets, with $k_{RUV} = 10$.

The raw antibody derived tag (ADT) counts matrix of each dataset is size-factor standardized and log-transformed using the *logNormCount* function from *scater*[39]. In scMerge2, we used all features as negative controls and used $k_{RUV} = 3$ in both levels in scMerge2-h.

**COVID-19 60k data collection (COVID-19 60k).** To evaluate the robustness of the parameters in scMerge2, we further created a smaller subset of data, which is derived from selecting the cells from moderate/mild patients of the Stephenson data from the COVID-19 200k data. The selected subset has 66,967 cells from 58 samples and 17,446 genes where the aim is to integrate three different batches in the Stephenson data.

**COVID-19 CyTOF data collection.** Two public COVID-19 PBMC CyTOF datasets (Supplementary Table 1) were downloaded from Flow-Repository with ID FR-FCM-Z2XA for Geanon data[40] (4,747,543 cells from 21 samples) and zenodo https://doi.org/10.5281/zenodo.6120249 for data from granulocyte depleted whole blood in COMBAT study[41] (7,118,158 cells from 160 samples), which both contain the expression matrix and cell type annotations. To combine the two studies, we manually unified antibody names and the cell type annotations to 18 cell types. The expression matrices were then used as input for scMerge2. Noted that we used all features as negative controls in scMerge2.

**COVID-19 IMC data collection.** The COVID-19 IMC dataset generated by[29] aims to assess the pathology of lungs across Covid-19 disease progression. The dataset, including cell intensities and metadata, was obtained from the repository https://zenodo.org/record/4139443#. Yw_gk9LMKXI provided in the publication and contained 237 images generated from 23 samples across 43 markers. In the original manuscript[29], the cell types were annotated by first clustering using the Leiden algorithm and then manually curated into 17 meta-clusters based on marker expression, phenotype, and proximity to lung structures.

### Evaluation

**Part I - Simulation.** Simulation framework. We adopted a simulation framework to generate single-cell multi-condition and multi-sample data with batch effect based on scDesign3[24]. This framework is able to simulate single-cell count data that preserve the gene-wise correlation structure. Similar to many other simulators, scDesign3 required a a real training scRNA-seq data to estimate the required parameters. Here, we have taken a subset of Stephenson data that contains four cell types (B cell, CD14 Monocytes, CD4 T and CD8 T) and 23 samples from two conditions (Healthy and Severe) as training data. From each sample, we randomly subsampled 400 cells. Only genes that were in the top 2000 highly variable genes and expressed in more than 2% of the cells were included. We further excluded any genes that were originally considered as differential expressed (with adjusted $p$-value < 0.2). This resulted in the training data with 9200 cells and 1196 genes from 23 samples. Our simulation framework includes three main steps.

Step 1: Construct a null dataset with no differentially expressed genes by first permuting the condition labels in the training data. We then estimate both cell-type and sample variation in the data using the function *fit_marginal()* in scDesign3 that fits the marginal distribution of each gene using a negative binomial distribution with the mu formula `~ cell type + sample ID + condition` and the sigma formula `~ 1`. Then we used a vine copula to estimate the gene correlation from the real training data.

Step 2: Introduce the batch effect to the simulated data. Assuming all genes are affected by the batch variation, we drew a vector with length equal to the number of genes from a log-normal distribution with mean log(2) and standard deviation 0.43 as batch effect on the mean of the gene distribution. The direction of the batch effect is randomly assigned to each gene.

Step 3: Introduce the ground truth differential state genes to the simulated data. For each cell type, we randomly select $p$% of genes to be differentially expressed between two conditions ($p = 5, 10$ in our study). The log fold changes (logFC) vector is simulated from a log-normal distribution, with the mean $\mu_{lfc}$ and the standard deviation $\sigma_{lfc}$. In our evaluation setting, we consider a range of logFC values from $\mu_{lfc} = 1.1$ to 2 in 0.1 increment and $\sigma_{lfc} = 0.43$. The direction of the regulation is randomly assigned to each DS genes using a binomial distribution with probability 0.5.

Lastly, with the fold change of both batch effect and condition effect combined with the parameters estimated in *Step 1*, the simulated single-cell data is generated from the negative binomial distribution using strategies implemented in *simu_new()* of scDesign3. For each value of logFC, we simulated 18,400 cells (23 samples, each sample with 800 cells), with 5% or 10% differential states genes within each cell types.

Evaluation metrics and settings—Differential states analysis. To assess the impact of data integration on downstream analytics, we considered the performance of the differential states analysis results on the simulated data. Our evaluation is based on three metrics; false discovery rate (FDR), true positive rate (TPR) and F1 scores. For each log-transformed simulated matrix with dimension $G \times C$, with $S$ samples and $T$ cell types, we took the gene-wise average of each sample within each cell type, resulting in a $G \times S$ matrix for each cell type. We then performed a differential state analysis using the limma-trend algorithm[25] on the cell-type specific sample-wise aggregated data using the default parameters.

**Part II - Real data comparison.** Evaluation setting for scRNA-seq and CITE-seq data collection.

1. Signal to noise ratio: We used ARI and ASW (see evaluation metrics below) to evaluate the concordance of clustering results with respect to the cell type labels and the datasets. A desirable data integration method will show a high concordance between the clustering result and known cell type information (signal refers to cell types) and a low concordance between the clustering results and known datasets information (noise refers to batch effect).

2. Severity prediction: We aggregated cell-type-specific average expression of each sample to a gene by sample matrix for each cell type. We then used each cell-type specific matrix to predict the sample condition (Healthy, Mild/Moderate and Severe/Critical) using support vector machine (SVM) with radial basis function kernel. The prediction performance was evaluated using repeated fivefold cross validation with 20 repeats. We evaluate the prediction performance using $F_1$ score.

3. Visualisation plot: For scRNA-seq data, we used Uniform Manifold Approximation and Projection (UMAP) to visualise and evaluate the results of the adjusted expression matrix. For CITE-seq case study, we used j-UMAP to jointly visualise the two modalities[32], where we first performed PCA within each modality, and then j-UMAP was performed to obtain the joint UMAP embeddings of the two modalities.

Evaluation on IMC data collection. We applied scMerge2 to perform data integration of the 23 samples. This is achieve by first filtering and selecting the data using the 38 markers specified in the original publication[29] and removing all undefined cell types (i.e. cells having cell type annotation as "nan"). Next, considering sample labels as batch information, we applied scMerge2 with settings $k_{RUV} = 2$, $k_{pseudoBulk} = 5$, $k_{celltype} = 20$, using all markers as negative control genes and highly variable genes. Thirdly, unsupervised clustering was performed on both the unnormalised and scMerge2 normalised datasets using the FlowSOM[30] algorithm with 17 clusters. The Adjusted Rand Index (ARI) was used to compared the concordance between this unsupervised

clustering with the manually curated cell types in the original manuscript[29]. The results are visualised using heatmaps showing the average marker abundance in the cell types. Average marker abundance were generated after scaling the marker expression by computing the ratio of the mean of each marker and its standard deviation.

Sensitivity analysis of scMerge2. We examined the robustness of the following parameters in scMerge2: the number of pseudo-bulk constructed; the number of neighbours in SNN graph; the pseudobulk construction strategy and the number of unwanted variation. We performed our sensitivity analysis on the COVID-19 60k data on a number of settings for each of the four parameters as below:

- Number of pseudobulk constructed within each group: 10, 20, 30, 40 and 50
- Number of neighbours in SNN graph: 5, 10, 15, 20, 25 and 30
- Ways of pseudobulk construction: Default, Pool-Divide, Pool-Divide (Random)
- Number of factors of unwanted variation to be removed: 10, 15, 20, 25 and 30

For each setting, we repeat the analysis 10 times with a different seed and assess the concordance performance of the signal to noise ratio using ASW and ARI as evaluation metrics as describe in the Section *Evaluation metrics*. We compared against benchmarking methods described in the Section *Benchmarking methods*.

**Evaluation metrics.** We used three metrics to assess the performance of data integration results from different methods. Details of the evaluation metrics are described as follows:

- Adjusted Rand Index (ARI) - Clustering analysis: We used ARI to quantify the concordance of the clustering results with respect to the cell type (ARI (cell type)) and batch labels (ARI (batch)). The clustering results for all methods were derived from first building a shared nearest neighbour from the batch corrected embeddings with a default number of neighbours of 10, followed by louvain clustering. For scMerge2, the batch corrected embeddings were derived from the top 20 PCs of the adjusted gene expression matrix.
- Average silhouette width (ASW) - Embedding visualisation: We calculated the average of silhouette coefficients for each cell (ASW) by considering two different groupings: cell type (ASW (cell type)) and batch label (ASW (batch)), based on the Euclidean distance obtained from the UMAP embeddings generated from the batch corrected embeddings.
- PCA scores: We calculated the coefficient of determination ($R^2$) for a linear regression model that fitted each of the first 20 principal component with technical variation labels, such as batch, technology and protocol labels. We then calculated the product of the variance explained by each principal component and the corresponding $R^2$. The final PCA score was calculated by summing across the products, which quantify how much the PCs explained the unwanted technical variation.

**Benchmarking methods.** We benchmarked the performance of scMerge2 against five other methods that are designed for data integration of scRNA-seq datasets in terms of the batch corrected embeddings in the COVID-19 200k data. Detailed settings used in each method are as follows:

(i) Seurat. Applying Seurat with canonical correlation analysis set as the reduction method. Version 4.1.1. of the *Seurat*[42] R package was used. We first identified the variable features within each batch using *FindVariableFeatures()* and then selected the integration features using *SelectIntegrationFeatures()*. The integration anchors were then identified using *FindIntegrationAnchors()* with reduction set as "cca", followed by *IntegrateData()* to obtain the integrated data.

(ii) SeuratRPCA. Similar to Seurat (CCA), within each batch, we first found the variable features, with an addition PCA step performed. After integration features were selected, *FindIntegrationAnchors()* was performed with reduction set as "rpca". Lastly, *IntegrateData()* was performed to obtain the integrated data.

(iii) fastMNN. This is a fast version of the mutual nearest neighbours (MNN) method[43]. R package *batchelor v1.12.3* was used. We ran *fastMNN()* with default parameters to derived both the batch corrected embeddings and adjusted expression matrix.

(iv) Liger. R package *rliger v1.0.0*[44] was used. Online integrative nonnegative matrix factorization was performed to obtain the batch corrected embedding following the tutorial (https://github.com/welchn-lab/liger/blob/master/vignettes/online_iNMF_tutorial.html), where we first ran *selectGenes()* to select the features, *scaleNotCenter()* to scale the features, and *online_iNMF()* with *miniBatch_size = 5000* and *max.epochs = 5*.

(v) Harmony. R package *Harmony v0.1.0*[18] was used. The PCA space returned by *runPCA()* of R package *scater* was used as input, and then *HarmonyMatrix()* was performed with *do_pca = FALSE* to retain the batch corrected embedding.

(vi) scVI. Python package *scvi v0.16.1*[19] was used. We used the following settings to perform the data integration using model *SCVI*: the number of layers as 2; the number of latent variables as 30 and the gene likelihood as negative binomial.

(vii) Scanorama. Python package *scanorama v1.7.3*[15] was used. We ran *correct_scanpy()* to perform the data integration and gene expression adjustment.

## Computational efficiency comparison

We used two of the COVID CITE-seq dataset (Liu and Stephenson) to evaluate the computational efficiency of the integration methods by varying the number of cells from 5k to 1M in four scenarios: (1) scRNA-seq: 17k genes with 2 batches; (2) scRNA-seq: 17k genes with 6 batches; (3) ADT: 87 proteins with two batches; (4) ADT: 87 proteins with six batches. The running time was measured using 1 core with the maximum number of threads of OpenMP library that can be used by a parallel region is set to 1. The peak resident set size, which is the highest amount of memory used by a process, was recorded to measure the memory usage. All methods were run using a research server equipped with dual Intel(R) Xeon(R) Gold 6148 Processor with 40 cores and 768 GB of memory and we set the running time limit as 10 hours. It is noted that for both Seurat and scMerge2 we evaluated two settings: one that adjusts the expression matrix for full gene matrix and another that only adjusts the top 2000 highly variable genes.

## COVID-19 downstream analysis

**Differential abundance analysis on the cells from mild/moderate and severe/critical samples.** Differential abundance (DA) analysis was performed on the cells from mild/moderate and severe/critical samples using DA-seq[26]. The top 30 PCs derived from the adjusted expression data were used as input for the algorithm to calculate the DA scores. A range of *k* values from 50 to 500 was used for the calculation of DA score vector with kNN. We define salient differential abundance (DA) cells as cells with absolute abundance scores >0.8.

**Differential states analysis of DA cells.** For all DA cells, we aggregated cell-type-specific abundance scores (or values) of each sample to a gene by sample matrix for each cell type. Next, we model the aggregated cell-type-specific abundance values across using a linear model with severity and the days since symptom onset as covariates. We account for sample level variability using the limma-trend implementation in the R package *limma*[25]. We then ranked the genes based on the test statistics. The preranked based gene set enrichment

analysis (GSEA) of the selected pathways that are related COVID-19 disease mechanism[28] (as listed in Fig. 4c) is measured using the *fgsea* function in the R package *fgsea v1.22.0*[45]. Significant pathways are defined with adjusted *p*-value <0.05.

## Statistics and reproducibility

All analysis was done in R version 4.1.2. No statistical method was used to predetermine sample size. No dataset listed in Supplementary Table 1 were excluded from the analyses. Cells with low-quality were excluded based on standard single-cell preprocessing procedures. The experiments were not randomized. The Investigators were not blinded to allocation during experiments and outcome assessment.

## Reporting summary

Further information on research design is available in the Nature Portfolio Reporting Summary linked to this article.

## Data availability

All data used in this study are included in Supplementary Table 1. **The COVID-19 IMC data.** The Arunachalam data used in this study is available in the GEO database under accession code GSE155673. The Bost_PBMC data is available in the GEO database under accession code GSE157344. The COMBAT data is available in the EGA database under accession code EGAS00001005493. The Combes data is available in the GEO database under accession code GSE163668. The Lee data is available in the GEO database under accession code GSE147507. The Liu data is available in the GEO database under accession code GSE161918. The Ramaswamy data is available in the GEO database under accession code GSE166489. The Ren data is available in the GEO database under accession code GSE158055. The Schulte-Schrepping data is available in the EGA database under accession code EGAS00001004571. The Schuurman data is available in the GEO database under accession code GSE164948. The Silvin data is available in the EBI database under accession code E-MTAB-9221. The Sinha data is available in the GEO database under accession code GSE157789. The Stephenson data is available in the EBI database under accession code E-MTAB-10026. The Su data is available in the EBI database under accession code E-MTAB-9357. The Thompson data is available in the GEO database under accession code GSE166992. The Unterman data is available in the GEO database under accession code GSE155224. The Wilk data is available in the GEO database under accession code GSE174072. The Yao data is available in the GEO database under accession code GSE154567. The Zhao data is available at Figshare [https://figshare.com/articles/dataset/seu_obj_h5ad/16922467]. The Zhu data is available at the CNGB database under project code CNP0001102. **The COVID-19 IMC data.** The COMBAT data used in this study is available in the EGA database under accession code EGAS00001005493. The Geanon data is available at FlowRepository under the accession code FR-FCM-Z2XA. **The COVID-19 IMC data.** The Rendeiro data is available in Zenodo under the accession code zenodo.4110560, zenodo.4139443 and zenodo.4637034 Source data are provided with this paper.

## Code availability

The code to run scMerge2 is part of the scMerge package [Github: https://github.com/SydneyBioX/scMerge][46].

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

## Acknowledgements

The authors thank all their colleagues, particularly at The University of Sydney, Sydney Precision Data Science Centre and Charles Perkins Centre for their support and intellectual engagement. A special thank to Mr. Dongyuan Song and Prof. Jingyi Jessica Li from UCLA for providing the R code for scDesign3. The following sources of funding for each author are gratefully acknowledged: the AIR@innoHK programme of the Innovation and Technology Commission of Hong Kong to J.Y.H.Y., E.P., Y.C. and Y.L. Australian Research Council Discovery Early Career Researcher Award (DE200100944) funded by the Australian Government to E.P.; Research Training Program Tuition Fee Offset and Stipend Scholarship and Chen Family Research Scholarship to Y.L.; Australian Government Research Training Program (RTP) Scholarship to Y.C. and Y.L.; and the University of Sydney Postgraduate Excellence Award for E.W. The funding sources had no impact on the study design; in the collection, analysis, and interpretation of data, in the writing of the manuscript, and in the decision to submit the manuscript for publication.

## Author contributions

J.Y.H.Y. and Y.L. conceived and designed the study with input from E.P. Y.L. and J.Y.H.Y. led the method development and guided the evaluation data analysis. Y.L. and Y.C. jointly curated the scRNA-seq data and Y.L. implemented all data analytics and developed the corresponding R code. E.W. and E.P. curated the IMC data and performed the data analytics related to imaging data. All authors wrote, read, reviewed the manuscript and approved the final version.

## Competing interests

The authors declare no competing interests.
