## [Peer Review File · Nature Communications]

Atlas-scale single-cell multi-sample multi-condition data integration using scMerge2Reviewer #1 (Remarks to the Author):

In this manuscript, Lin et al. propose a major update to their method for single-cell data integration called scMerge2 that makes a number of improvements over the previous method to better enable large-scale, many-sample integration. The main advances here are a hierarchical integration approach that utilizes pseudo-bulk for scalability and pseudo-replication for removing unwanted variability. They demonstrate the method on a large collection of COVID-19 PMBC samples and show improved performance in terms of several downstream analytical tasks. Overall, this work presents a fairly performant integration method for single cell data but there are a few areas where improvements could be made. I have the following specific comments:

Main comments:

1. Given that the method is called scMerge2, it would be nice to see comparisons in some of the benchmarks to the initial implementation in scMerge (assuming that it is able to run on the benchmark datasets). This would also help highlight the necessity of the update.
2. Hierarchical integration is one of the three key innovations put forth by scMerge2. However, scMerge2 isn't the only method that offers this feature. Both fastMNN (via the merge.order parameter) and Seurat (via the sample.tree parameter) enable this functionality. For comparison purposes, the same hierarchical structure of merging/integration should be applied when comparing to methods that offer that functionality.
3. One of the major selling points of scMerge2 is that it can return fully corrected gene expression matrices as opposed to just integrated low dimensional embeddings. One aspect that is explored here is improvements to differential expression (or differential state) analysis and the results are compared to Seurat and fastMNN. However, neither of these other software packages recommend using corrected expression data for differential statistical analyses as the correction procedure introduces cell-cell dependencies that invalidate the assumptions of the downstream statistical tests. Is there a similar concern about dependency introduction with scMerge2? An alternate common workflow is to use the integrated low dimensional embeddings to define cell types and then run differential tests using the uncorrected data with a model that allows for covariates like batch, sample, etc. How does scMerge2 compare on these differential expression based metrics if this strategy is used for the other methods?
4. One feature that makes this method attractive is the purported scalability. It would be nice to see how both time and memory requirements increase with dataset size.

Minor comments:

1. There is quite a bit of background given on merging strategies in the third paragraph of the discussion. I think much of this could be better placed earlier in the article, with details in the methods section.
2. In Figure 3a, in the UMAP where cells are colored by study, it is a bit hard to tell how well mixed the studies are since the yellow color is plotted over top of everything. Sometimes, randomizing the plotting order of the points can make this easier to see, though the supplementary figure does help with this as well. It might also be worth showing an unintegrated version of this dataset (similar to the CyTOF example).
3. The color choice for Figure 4 panel D makes it difficult to distinguish the two states. I would recommend changing to more contrasting colors.

Reviewer #2 (Remarks to the Author):

The authors present scMerge2 that is an algorithm for integration of single cell data across samples, conditions and studies. scMerge2 is an improvement to their previous tool, scMerge that has been published previously (<https://www.pnas.org/doi/10.1073/pnas.1820006116>). It seems that the biggest change between scMerge and scMerge2 is the addition of hierarchical integration

and pseudo-bulk construction.

scMerge2 works on data from different single cell technologies. The authors demonstrate tests on CYTOF and CITE-Seq. The algorithm is tested on a large example that uses a COVID-19 data collection and the results recapitulate existing knowledge on increase of certain cell types with disease severity, such as macrophages. I agree with the increased need for such methods to integrate atlases across individuals and conditions. This is currently a very active area of research with lots of tools available.

The performance of scMerge2 is compared against five different batch correction/integration tools. However, I do wonder how scMerge2 performance is improved from scMerge and also how its performance compares to two of the top performing single cell integration methods scVI and scanorama in a benchmarking study that was published earlier in the year. The authors do not test scMerge2 against those top performing tools.

Reviewer #3 (Remarks to the Author):

In this work, the authors developed scMerge2, which is a framework based on factor analysis model that integrates scRNA-Seq datasets from multiple samples and multiple biological conditions. scMerge2 was also implemented to integrate multiple CITE-Seq datasets, where gene expression and epitopes are jointly profiled. To enable more efficient computation, scMerge2 leverages pseudo-bulk by aggregating cells from similar cell types. The authors demonstrated that scMerge2 can be applied to atlas-level data with millions of cells. Another key feature of scMerge2 is that it can obtain the adjusted gene expression matrix (instead of merely the low-dimensional embeddings), which can be used for gene-level downstream analysis. This is an interesting work. The followings are my comments:

1. The authors demonstrated that scMerge2 can be implemented on atlas-level datasets. It would be great if the memory cost and CPU time can also be reported, when the number of cells varies.
2. k-nearest neighbor batch-effect test (kBET) is a popular criterion to evaluate the performance of batch effect correction for data integration. It would be great if the authors can add this criterion in the comparison.

Reference for kBET:

Zhichao Miao, F Alexander Wolf, Sarah A Teichmann, and Fabian J Theis. A test metric for assessing single-cell RNA-seq batch correction. *Nature Methods*, 16(1):43-49, 2019.

3. In scMerge2, a set of single-cell stable genes was used to guide the estimate of the factor analysis model. How is this set of genes selected, is it the same for all the datasets?
4. To broaden the impact of scMerge2, it would be interesting to demonstrate the integration results across different platforms, for example, the integration of smart-seq datasets with 10X Visium datasets. And maybe also cross-species integration?
5. scMerge2 aggregates cells from the same cell type identified by Louvain clustering. It would be great if the authors can comment on the case when there is a developmental lineage of cells, will it affect the performance of scMerge2?

REVIEWER COMMENTS

Reviewer #1 (Remarks to the Author):

In this manuscript, Lin et al. propose a major update to their method for single-cell data integration called scMerge2 that makes a number of improvements over the previous method to better enable large-scale, many-sample integration. The main advances here are a hierarchical integration approach that utilizes pseudo-bulk for scalability and pseudo-replication for removing unwanted variability. They demonstrate the method on a large collection of COVID-19 PMBC samples and show improved performance in terms of several downstream analytical tasks. Overall, this work presents a fairly performant integration method for single cell data but there are a few areas where improvements could be made. I have the following specific comments:

Response: We thank the reviewer for the positive feedback and we have addressed each raised comment in detail below.

Main comments:

1. Given that the method is called scMerge2, it would be nice to see comparisons in some of the benchmarks to the initial implementation in scMerge (assuming that it is able to run on the benchmark datasets). This would also help highlight the necessity of the update.

Response: We appreciate the reviewer's suggestion. As suggested, we performed the comparison between scMerge and scMerge2, in terms of integration performance as well as its computational usage. Due to the intensive computation resource demand, we only benchmarked the integration performance of COVID60k data. As shown in Figure R1, scMerge2 outperforms its previous version in terms of both integration and computational performance. Notably, we have significantly improved the computational efficiency of the method, which now requires only ~3 minutes ~20Gb for 50k cells with 17k genes, compared to over 4 hours and 50 Gb in the previous version (**Figure R1c-d**).

Figure R1 (a-b) Data integration evaluation: Scatter plots of evaluation metrics of data integration of a 60k cells subset of one COVID-19 study (Stephenson) for scMerge, scMerge2, Seurat, Seurat (RPCA), Harmony, fastMNN, Liger and Raw: (a) Adjusted rand index (ARI), where x-axis indicates 1 minus batch ARI and y-axis indicates cell type ARI; (b) Average silhouette width (ASW), where the x-axis is 1 minus batch ASW and y-axis is the cell type ASW. (c-d) Computational performance evaluation: (c) computational time (min) for each method increases with the number of cells in the data; (d) memory usage (Gb) for each method increases with the number of cells.

2. Hierarchical integration is one of the three key innovations put forth by scMerge2. However, scMerge2 isn't the only method that offers this feature. Both fastMNN (via the merge.order parameter) and Seurat (via the sample.tree parameter) enable this functionality. For comparison purposes, the same hierarchical structure of merging/integration should be applied when comparing to methods that offer that functionality.

Response: We thank the reviewer for this comment. It is important to note that both fastMNN and Seurat only allow pairwise merging each time and merge multiple batches in a progressive way. This is distinct to the innovation in scMerge2h, which can take a flexible multi-level merging structure which allows merging of multiple batches at one time on one level. For example, for a data integration task (COVID-19 200k), where we have batch 1, 2, 3 from dataset Liu and batch 4, 5, 6 from dataset Stephenson, we want to remove the batch effect within each dataset and then merge across two datasets. Both fastMNN and Seurat need to merge batch 1 with 2, then

1+2 with 3; batch 4 with 5, and then 4+5 with 6; and finally, 1+2+3 with 4+5+6, while scMerge2h can merge batch 1, 2, 3 together; batch 4, 5, 6 together; and finally merge 1+2+3 and 4+5+6.

That being said, we have followed the suggestion and performed the comparison of scMerge2-h with fastMNN (merge.order) and Seurat (sample.tree with user specified merging order as required). The results is shown in **Figure R2**, we find that among the three methods that allow hierarchical merging, the overall performance of scMerge2h outperforms the other two methods (Also see **Figure R6c** in Reviewer #2 comment 2 for the overall rankings). It is noted that specifying the order of Seurat and fastMNN do not necessarily perform better than its default settings, which will decide the merging order automatically. We have added the results and discussion of these two methods in our results section of the manuscript (See **Line 100-105** in Page 4 for discussion, **Line 126 - 129** in Page 5 and **Supplementary Fig. S1** for results).

Figure R2 (Supplementary Fig. S1) Data integration evaluation to compared methods with hierarchical merging settings: Scatter plots of evaluation metrics of data integration of a 200k cells COVID-19 data for scMerge2, scMerge2-h, SeuratRPCA, SeuratRPCA-h, fastMNN and fastMNN-h: (a) Adjusted rand index (ARI) , where x-axis indicates 1 minus batch ARI and y-axis indicates cell type ARI; (b) Average silhouette width (ASW), where the x-axis is 1 minus batch ASW and y-axis is the cell type ASW.

3. One of the major selling points of scMerge2 is that it can return fully corrected gene expression matrices as opposed to just integrated low dimensional embeddings. One aspect that is explored here is improvements to differential expression (or differential state) analysis and the results are compared to Seurat and fastMNN. However, neither of these other software packages recommend using corrected expression data for differential statistical analyses as the correction procedure introduces cell-cell dependencies that invalidate the assumptions of the downstream statistical tests. Is there a similar concern about dependency introduction with scMerge2? An alternate common workflow is to use the integrated low dimensional embeddings to define cell types and then run differential tests using the uncorrected data with a model that allows for covariates like batch, sample, etc. How does scMerge2 compare on these differential expression based metrics if this strategy is used for the other methods?

Response:

We thank the reviewer for the insightful comments and suggestions. Following the recommendation of the reviewer, we performed additional comparisons with the following settings in the differential state (DS) analysis:

- (1) uncorrected data with batch information as covariates (logcounts (batch))
- (2) scMerge2 normalized data with batch information as covariates (scMerge2 (batch))
- (3) scMerge2 normalized data with batch information as covariates, and force the original zeros as zeros (scMerge2 (batch + binary))
- (4) Seurat normalized data with batch information as covariates (Seurat (batch))
- (5) Seurat normalized data with batch information as covariates, and force the original zeros as zeros (Seurat (batch + binary))

We used the same simulated data we used in **Figure 2c** and **Supplementary Fig. S3-S5**. In particular, we focused on the scenario when the threshold of the p-value is 0.05, the percentage of DS genes in the simulated data is 5%, and we used trend limma on the aggregated bulk data.

As **Figure R3** shows, we find that scMerge2 performed comparably to uncorrected data with batch information as covariates (logcounts (batch)) in terms of F1, while exhibiting a more stable performance in controlling FDR. We also explore the performance of incorporating the batch information in scMerge2 and Seurat normalized data. We find that incorporating the batch covariate in scMerge2 normalized data (scMerge2 (batch) and scMerge2 (batch + binary)) in general further improves the stability and performance in terms of TPR and F1, although it slightly increases FDR. We note that all three settings of scMerge2 outperformed the three settings in Seurat.

In addition, we would like to highlight that our evaluation of other downstream analytics, such as prediction modeling, revealed that scMerge2 was able to better predict the outcome of interest in the COVID dataset, as demonstrated in **Figure 3c**. Despite the potential dependency introduced by scMerge2, we did not observe any adverse impact on downstream analysis. Moreover, as shown in **Figure 4**, the ability of scMerge2 to provide a fully corrected gene expression matrix significantly enhances the flexibility of downstream analysis, including gene set enrichment analysis and cell-cell interaction analysis.

Together, we believe scMerge2 provides a more reliable normalized matrix compared to the other existing methods which can be used to enhance diverse downstream analysis.

Figure R3 Performance of DS analysis with additional methods comparison on the scenario when the threshold of the p-value is 0.05, the percentage of DS genes in the simulated data is 5%, with varying mean logFC of DS genes. The row shows three evaluation metrics: F1, TPR and FDR and the column indicates 4 cell types in the simulated data.

4. One feature that makes this method attractive is the purported scalability. It would be nice to see how both time and memory requirements increase with dataset size.

Response: We appreciate the reviewer's comment. As suggested, we performed additional experiments to evaluate the time and memory requirements as the dataset size increases. Using two of the COVID CITE-seq dataset (Liu and Stephenson), which in total have ~1M cells, we benchmarked the scalability of the integration methods by varying the number of cells from 5k to 1M in four scenarios:

1. scRNA-seq - ~17k genes with 2 batches;
2. scRNA-seq - ~17k genes with 6 batches;
3. ADT - 87 proteins with 2 batches;
4. ADT - 87 proteins with 6 batches.

The running time was measured using 1 core with the maximum number of threads of OpenMP library that can be used by a parallel region is set to 1. The peak resident set size, which is the highest amount of memory used by a process, was recorded to measure the memory usage. All methods were run using a research server equipped with dual Intel(R) Xeon(R) Gold 6148 Processor with 40 cores and 768 GB of memory and we set the runtime limit as 10 hours. We performed evaluation on all eight methods. In particular, for Seurat and scMerge2, we evaluated

these methods under two settings: one that adjusts the expression matrix for full gene matrix and another that only adjusts the top 2000 highly variable genes.

We find that among all the methods that are able to return the adjusted gene expression matrix, scMerge2 is the most efficient in terms of computational time in all comparison (**Figure R4-R5**), requiring around ~1 hour to integrate 1 million of cells with 17k genes and ~25 min to integrate 1 million cells with 87 proteins. In terms of memory usage, to return a full adjusted gene expression matrix, scMerge2 requires more memory than fastMNN and scVI, but significantly less than Seurat, as Seurat (RPCA) encountered a memory error when integrating the fully gene expression matrix of 500k cells. We suggest that users consider returning a subset of genes, such as highly variable genes, when computational resources are limited. This can be done by setting `return_subset = TRUE` and specifying the gene list in `return_subset_genes` in `scMerge2()`. We have included these results in our manuscript (See Page 5, **Line 131 - 138**, **Supplementary Fig. S2-S3**, and Method Section Computational efficiency comparison).

Figure R4 (Supplementary Fig. S2) Scalability of each data integration method of the RNA integration task with (a-b) 2 batches in the data and (c-d) 6 batches in the data, colored by the methods and the type of the lines indicate the output type of the methods. (a, c) shows

computational time (min) for each method increase with the number of cells in the data; (b, d) shows memory usage (Gb) for each method increase with the number of cells. Note that SeuratRPCA encountered a memory error when integrating the full gene expression matrix of 500k cells, and SeuratRPCA (hvg) encountered a memory error when integrating the full gene expression matrix of 1 million cells for 6 batches cases.

Figure R5 (Supplementary Fig. S3) Scalability of each data integration method of the ADT integration task with (a-b) 2 batches in the data and (c-d) 6 batches in the data, colored by the methods and the type of the lines indicate the output type of the methods. (a, c) shows computational time (min) for each method increase with the number of cells in the data; (b, d) shows memory usage (Gb) for each method increase with the number of cells.

Minor comments:

1. There is quite a bit of background given on merging strategies in the third paragraph of the discussion. I think much of this could be better placed earlier in the article, with details in the methods section.

Response: We thank the reviewer for the suggestions. We included and expanded the method description of the merging strategies in the method section (**Line 425-429** in Page 15) and added the reference to the method section in our results section. We have further highlighted the flexibility of our hierarchical merging strategies compared to the other two methods in the results section, as also discussed in the comment 2 above (See **Line 100-105** in Page 4). As our discussion currently mainly focuses on the variety of hierarchical merging strategies, we believe it should stay in the discussion.

2. In Figure 3a, in the UMAP where cells are colored by study, it is a bit hard to tell how well mixed the studies are since the yellow color is plotted over top of everything. Sometimes, randomizing the plotting order of the points can make this easier to see, though the supplementary figure does help with this as well. It might also be worth showing an unintegrated version of this dataset (similar to the CyTOF example).

Response: We appreciate the reviewer for the suggestion. We agree that with the ~5 million points, it is very hard to visualize the data effectively. The current plot in **Figure 3a** is already with points randomized, and therefore, we also put in **Supplementary Fig. S9** which overlays each dataset on the full UMAP coordinates to help with the visualization, as the reviewer pointed out.

3. The color choice for Figure 4 panel D makes it difficult to distinguish the two states. I would recommend changing to more contrasting colors.

Response: Thank you for your comment. We acknowledge that the color choice used in **Figure 4D** may not be ideal. However, we made a conscious decision to maintain consistency with the color legend in **Supplementary Fig. S8**, which resulted in a trade-off between consistency and clarity during the color selection process.

Reviewer #2 (Remarks to the Author):

The authors present scMerge2 that is an algorithm for integration of single cell data across samples, conditions and studies. scMerge2 is an improvement to their previous tool, scMerge that has been published previously (<https://www.pnas.org/doi/10.1073/pnas.1820006116>). It seems that the biggest change between scMerge and scMerge2 is the addition of hierarchical integration and pseudo-bulk construction.

scMerge2 works on data from different single cell technologies. The authors demonstrate tests on CYTOF and CITE-Seq. The algorithm is tested on a large example that uses a COVID-19 data collection and the results recapitulate existing knowledge on increase of certain cell types with disease severity, such as macrophages. I agree with the increased need for such methods to integrate atlases across individuals and conditions. This is currently a very active area of research with lots of tools available.

The performance of scMerge2 is compared against five different batch correction/integration tools. However, I do wonder how scMerge2 performance is improved from scMerge and

Response: We appreciate the reviewer’s suggestions. As suggested, we conducted a comparison of scMerge and scMerge2 in regards to their integration performance and computational usage. Please see our response to Reviewer #1 comment 1 and **Figure R1** for details. In summary, we find that scMerge2 represents a significant improvement over the previous version of scMerge, as it enhances both computational efficiency and data integration performance.

also how its performance compares to two of the top performing single cell integration methods scVI and scanorama in a benchmarking study that was published earlier in the year. The authors do not test scMerge2 against those top performing tools.

Response: We appreciate the reviewer’s comment. As suggested, we have added scVI and Scanorama in our benchmarking. **Figure R6** demonstrates that scMerge2 exhibits superior data integration performance in comparison to these two methods. We have updated **Figure 2** with the benchmarking results of the two additional methods. It is worth noting that as scMerge2 is also much more computationally efficient than both scVI and Scanorama in terms of running time. Additionally, when adjusting the full gene expression matrix, scMerge2 utilizes notably less memory than Scanorama and performs comparably to scVI when adjusting for top highly variable genes. For more details of computational usage benchmarking, please refer to Reviewer #1 comment 4 and **Figure R4-R5**.

Figure R6 (a) Scatter plots of evaluation metrics of data integration of a 200k cells subset of two COVID-19 studies (Liu and Stephenson) for scMerge2, scMerge2-h (data merged in a hierarchical manner), Seurat, Seurat (RPCA), Harmony, fastMNN, Liger, scVI, Scanorama and Raw: Adjusted rand index (ARI) (left panel), where x-axis indicates 1 minus batch ARI and y-axis indicates cell type ARI; Average silhouette width (ASW), where the x-axis is 1 minus batch ASW and y-axis is the cell type ASW (right panel). (b) Dot plots indicate the ranking of the data integration methods in terms of five different evaluation metrics. The size of the dot indicates the scaled scores, which are obtained from the min-max scaling of the original values. The overall ranking is ranked based on the average ranking of the five evaluation metrics.

Reviewer #3 (Remarks to the Author):

In this work, the authors developed scMerge2, which is a framework based on factor analysis model that integrates scRNA-Seq datasets from multiple samples and multiple biological conditions. scMerge2 was also implemented to integrate multiple CITE-Seq datasets, where gene expression and epitopes are jointly profiled. To enable more efficient computation, scMerge2 leverages pseudo-bulk by aggregating cells from similar cell types. The authors demonstrated that scMerge2 can be applied to atlas-level data with millions of cells. Another key feature of scMerge2 is that it can obtain the adjusted gene expression matrix (instead of merely the low-dimensional embeddings), which can be used for gene-level downstream analysis. This is an interesting work. The followings are my comments:

Response: We thank the reviewer for the positive feedback and we have provided a detailed response to each raised comment below.

1. The authors demonstrated that scMerge2 can be implemented on atlas-level datasets. It would be great if the memory cost and CPU time can also be reported, when the number of cells varies.

Response: In response to the suggestion, we conducted further experiments to assess how the time and memory requirements change with an increase in the dataset size. Please see our response to Reviewer #1 comment 4 and **Figure R4-R5** for details. Overall, we find that our method outperforms the other methods that can return a full adjusted gene expression matrix, with faster computational time and similar performance in terms of memory cost.

2. k-nearest neighbor batch-effect test (kBET) is a popular criterion to evaluate the performance of batch effect correction for data integration. It would be great if the authors can add this criterion in the comparison.

Reference for kBET:

Zhichao Miao, F Alexander Wolf, Sarah A Teichmann, and Fabian J Theis. A test metric for assessing single-cell RNA-seq batch correction. *Nature Methods*, 16(1):43-49, 2019.

Response: We appreciate the reviewer's suggestions. While we acknowledge the importance of this popular metric in the field, we find that in our benchmark studies, this metric is very sensitive to batch effects as well as the sample size of the data. This is due to the fact that the metric is based on Chi-squared statistics, which are known to be sensitive to sample size. This issue has also been discussed in one of the Github issues of kBET (<https://github.com/theislab/kBET/issues/47>), where the maintainer commented that "We are aware that kBET is a sensitive tool and often reports rejection rates close to 1." This is consistent with what **Figure R7** below shows, where we subsampled 0.1~20% of cells from the COVID200k data and calculated kBET metrics. We find that the metric is very sensitive to the number of cells used as input and all methods report rejection rates close to 1 when equal or more than 10k cells (only 5% of the data) are subsampled from the data. This makes the performance of the data integration methods indistinguishable. As a result, we made the decision not to include this metric in our benchmarking.

Figure R7 Boxplots indicates the kBET observed value for each method using different proportions of cells (0.1% ~ 20%) in the data.

3. In scMerge2, a set of single-cell stable genes was used to guide the estimate of the factor analysis model. How is this set of genes selected, is it the same for all the datasets?

Response: We appreciate the reviewer's comment. This set of gene is a predefined gene list that was defined from our previous studies (Lin, Ghazanfar, Strbenac, *et al.*, 2019; Lin, Ghazanfar, Wang, *et al.*, 2019), where we selected the genes with minimal association with cell types and developmental stages from a human early development single-cell dataset. Similar to Fig 2C in scMerge paper, we performed additional experiments to investigate scMerge2 performance using different lists of stably expressed genes, where we benchmarked our stably expressed genes against three choices of negative controls. These are (1) all genes; (2) all non highly variable genes with insignificant p-value; (3) top non highly variable genes based on the p-value. As **Figure R8** shows, We find that in general, scMerge2 performance is stable against the choice of negative control, with using stably expressed genes performs relatively better than the other choice of negative control genes.

Figure R8 Robustness analysis of the choice of negative control list used in scMerge2 using COVID-19 60k data: Adjusted rand index (ARI) (left panel), where x-axis indicates 1 minus batch ARI and y-axis indicates cell type ARI; Average silhouette width (ASW), where x-axis indicates 1 minus batch ASW and y-axis indicates cell type ASW (right panel).

4. To broaden the impact of scMerge2, it would be interesting to demonstrate the integration results across different platforms, for example, the integration of smart-seq datasets with 10X Visium datasets. And maybe also cross-species integration?

Response: We thank the reviewer's comment. As suggested, we demonstrate scMerge2 capacity in integrating data from different platforms and cross-species.

Different platforms

We integrated a spatially resolved dataset generated from MERFISH from mouse primary motor cortex (MOp) (Zhang *et al.*, 2021) with a scRNA-seq dataset generated by SMARTER platform (Yao *et al.*, 2021). We combine the two datasets via the 250 common genes shared between the two platforms. It is also notable that these two datasets have the following properties:

(1) Unbalanced number of cells: MERFISH data has ~280k cells and SMARTer data has ~6.3k cells

(2) Distinguished cell type proportions: MERFISH data has 8 cell types uniquely presented ("L4/5 IT", "L5/6 NP", "L6 IT Car3", "Micro", "Oligo", "OPC", "Peri" and "PVM") (Figure R8 a-b).

As **Figure R9** shows, scMerge2 is capable of removing the batch effect between these two very different platforms and at the same time reveal the unique cell types in each platform.

Figure R9 mouse primary motor cortex (MOp): (a-b) cell type proportion of the MERFISH and SMARTer data.(c) UMAP of raw and scMerge2.

Cross-species

We integrated single-nucleus RNA-seq (snRNA-seq) of adult hippocampal-entorhinal cells from dentate gyrus (DG) region in human (*Homo sapiens*), macaque (*Macaca mulatta*), and pig (*Sus scrofa*) (Franjic *et al.*, 2022). We concatenated the gene expression matrices across species based on the one-to-one gene homology mapping strategy based on ENSEMBL. As **Figure R10** shows, scMerge2 is able to remove the species effect in the dataset and also reveal the homologue cell types, such as neuroblasts (NB) that only exist in macaque and pig, but not human.

Figure R10 UMAP plots of cross species data integration.

5. scMerge2 aggregates cells from the same cell type identified by Louvain clustering. It would be great if the authors can comment on the case when there is a **developmental lineage** of cells, will it affect the performance of scMerge2?

Response: We thank the reviewer's comment. For data with continuous biological processes, one parameter that could be optimized is the number of neighbors in the shared nearest neighbor graph. A smaller of this parameter can result in better capture of local and detailed

variation. We demonstrated the capability of scMerge2 in integrating data with a developmental lineage of cells. Specifically, we integrated a scRNA-seq dataset from human hematopoiesis, consisting of 6 samples from CD34 + BMMC, BMMC and PBMC (Granja *et al.*, 2019), with a smaller number of neighbors set (specified by $k_{knn} = 5$). As **Figure R11** shows, scMerge2 effectively removes the batch effect in the dataset, revealing both the leukemic developmental trajectory and distinct cell types from different sample sources.

Figure R11 UMAP plots of human hematopoiesis data integration.

Reference

Franjic, D. *et al.* (2022) 'Transcriptomic taxonomy and neurogenic trajectories of adult human, macaque, and pig hippocampal and entorhinal cells', *Neuron*, 110(3), pp. 452–469.e14.

Granja, J.M. *et al.* (2019) 'Single-cell multiomic analysis identifies regulatory programs in mixed-phenotype acute leukemia', *Nature biotechnology*, 37(12), pp. 1458–1465.

Lin, Y., Ghazanfar, S., Strbenac, D., *et al.* (2019) 'Evaluating stably expressed genes in single cells', *GigaScience*, 8(9). Available at: <https://doi.org/10.1093/gigascience/giz106>.

Lin, Y., Ghazanfar, S., Wang, K.Y.X., *et al.* (2019) 'scMerge leverages factor analysis, stable expression, and pseudoreplication to merge multiple single-cell RNA-seq datasets', *Proceedings of the National Academy of Sciences of the United States of America*, 116(20), pp. 9775–9784.

Yao, Z. *et al.* (2021) 'A transcriptomic and epigenomic cell atlas of the mouse primary motor cortex', *Nature*, 598(7879), pp. 103–110.

Zhang, M. *et al.* (2021) 'Spatially resolved cell atlas of the mouse primary motor cortex by MERFISH', *Nature*, 598(7879), pp. 137–143.

Reviewer #1 (Remarks to the Author):

I appreciate the authors' work to address my questions and concerns. All major concerns have been addressed and I can recommend publication.

Reviewer #3 (Remarks to the Author):

All my previous comments have been reasonably addressed. I do not have any further comments.